# Western red cedar (*Thuja plicata*) beehives have no impact on honey bee (*Apis mellifera*) overwintering colony survival or detoxification enzyme expression

Alison McAfee[1,2]*, David R. Tarpy[2], Leonard J. Foster[1,3]*

1 Michael Smith Laboratories, Department of Biochemistry and Molecular Biology, University of British Columbia, Vancouver, British Columbia, Canada, 2 Department of Applied Ecology, North Carolina State University, Raleigh, North Carolina, United States of America, 3 Life Sciences Institute, Department of Biochemistry and Molecular Biology, University of British Columbia, Vancouver, British Columbia, Canada

* alison.n.mcafee@gmail.com (AM); foster@msl.ubc.ca (LJF)

## Abstract

In North America, wooden honey bee hives are most often constructed from pine, but some companies also produce and sell boxes made of western red cedar (*Thuja plicata*) as a result of its local availability and desirable properties. However, there is debate within the beekeeping community about whether cedar is a safe hive material for bees, since resins within the wood are known to be insecticidal or insect deterrents. There is very little empirical evidence to support or refute these arguments. Here, we recorded health metrics of honey bee nucleus colonies hived in western red cedar and pine boxes (n = 10 each) to determine if the type of wood affects colony outcomes. Colonies were produced and introduced into these boxes in late July, with monitoring continued until the following spring. We found no significant differences in adult bee populations, brood areas, or *Varroa* mite prevalence among colonies hived in cedar versus pine boxes at either the end of summer (September 1st) or spring (April 1st) assessments. Overwintering survival was identical in the two groups at 90%. Hemolymph detoxification enzyme expression differed strongly between callow (day-old) workers and foragers but did not differ with hive material. Overall, we did not find evidence that hiving honey bee colonies in boxes constructed of western red cedar had any negative or positive effect on bee physiology or colony outcomes.

## Introduction

Honey bee (*Apis mellifera*) hive construction material has varied through history, from hollowed logs or clay cylinders, to the straw skep, to the polystyrene boxes occasionally used today [1–3]. The most common material for building contemporary bee boxes, though, is wood, the majority of which is pine in North America (white pine or ponderosa pine, according to the websites of major suppliers), where it has been used since at least the 1800s [4]. Although less common, cedar has also been used for beehives over a similar time frame [4]. Cedar hives are mainly marketed to hobby beekeepers or used in situations where artisan products are valued [5], but they are also occasionally used by commercial beekeepers.

**Data availability statement:** All mass spectrometry raw files, sample metadata, label-free quantitation results, and the FASTA search database are available on the MassIVE (https://massive.ucsd.edu/ProteoSAFe/static/massive.jsp) proteomics data archive under the accession MSV000093767. All other data underlying the conclusions of this manuscript are available as supplementary information (S1 File and S2 File).

**Funding:** This work was supported by a grant from the Canadian Bee Research Fund to A.M. and L.J.F. (no applicable grant number) and a grant from the Natural Sciences and Engineering Research Council (RGPIN-2022-03022) to L.J.F. Mass spectrometry infrastructure used here was supported by grants from PacifiCan (grant number 22637), the Canada Foundation for Innovation and the BC Knowledge Development Fund (grant number 43403 for both), and the University of British Columbia Life Sciences Institute (no applicable grant number). The funders had no role in study design, data collection and analysis, decision to publish, or preparation of the manuscript.

**Competing interests:** The authors have declared that no competing interests exist.

"Cedar" is an ambiguous term that can colloquially refer to many species of coniferous tree, including the western red cedar (*Thuja plicata*), eastern red cedar (*Juniperus virginiana*), yellow cedar (*Chamaecyparis nootkatensis*), northern white cedar (*Thuja occidentalis*), Atlantic white cedar (*Chamaecyparis thyoides*), Port Orford cedar (*Chamaecyparis lawsoniana*), and incense cedar (*Calocedrus decurrens*), in addition to any of the four "true cedars" in the *Cedrus* genus (*i.e.*, *Cedrus atlantica, Cedrus brevifolia, Cedrus libani*, and *Cedrus deodara*, the former three of which are native to regions surrounding the Mediterranean Sea and the latter of which is native to the Himalayas) [6]. These species generally share the feature of containing pesticidal compounds in their resin; however, the activity of these compounds differs between tree species, and different insect and arachnid pests have different susceptibilities [7–12].

Western red cedar is the variety most commonly harvested for lumber in North America's Pacific Northwest region. While it is typically more expensive than pine, western red cedar has the desirable properties of being lightweight [13] and naturally rot-resistant [14], while also being more insulating and less prone to warping than pine [15,16]. However, one of its most valued features – its pesticidal properties [16] – could conceivably make western red cedar an inappropriate material for hiving honey bees, and this possibility needs formal testing. While some research has compared outcomes of bees in "cedar" hives versus polyurethane foam [17], the exact species of tree from which the lumber was derived in these observations was not described, nor was a comparison to other woods conducted. To the best of our knowledge, research comparing outcomes of colonies housed in western red cedar hives has not yet been compared to conventional pine boxes.

Volatile and non-volatile bioactive compounds (extractives) are present in western red cedar wood, and those with confirmed pesticidal properties are found within the volatile fraction [16]. These include thujic acid, thujic methyl ester, methyl thujate, β-thujaplicin, and γ-thujaplicin, of which the former two are predominantly fungicidal and the latter three are predominantly insecticidal [18–24]. Insecticidal activity has been demonstrated against several beetle species, one moth, and one termite [16,21,23,25]. The exact mechanism of insecticidal activity has not been determined, but methyl thujate, β-thujaplicin, and γ-thujaplicin are monoterpenes; therefore, they may be acting as acetylcholinesterase inhibitors, as has been determined for other monoterpenes [22].

One way that insects, including honey bees, mitigate effects of toxins is metabolic resistance via detoxification enzyme activity [26] – a topic that has received much attention given honey bees' exposure to agricultural pesticides [27]. Honey bees possess all major classes of detoxification enzymes, including glutathione S-transferases (GSTs), carboxylesterases (COEs), and cytochrome p450 monooxygenases (CYP450s) [28,29]. Although honey bees do not feed directly on plant foliage, roots, or stems (which may have higher levels of toxins or insect deterrents) and thus have limited evolutionary exposure to xenobiotic compounds, their general sensitivity to insecticides appears to be comparable to that of other insects [30]. Honey bees do, however, gather tree resins that are used as a construction material (propolis) in the hive, but there is little data to suggest that they forage on species commonly known as cedar [31,32]. Activity of western red cedar extractives against honey bees is not yet known.

A honey bee colony is typically made up of a single reproductive queen, tens of thousands of (normally) sterile female workers, and a smaller fraction (up to ~5–10%) of seasonal reproductive males (drones) [33]. Honey bee larvae and pupae, which develop in cells constructed of wax, do not have an opportunity to contact wooden components of the hive until adulthood. During the summer, adult workers live for an average of 3–5 weeks [34], during which time they transition from working on in-hive tasks to foraging outside the colony [33]. As adults, the bees could presumably become exposed to compounds within or on the walls of their hive if those compounds are sufficiently volatile to vaporize into the air or if workers

physically contact resins. During the winter, adult workers live much longer (upwards of 6 months) [35] with little foraging activity outside the colony; therefore, hypothetical exposure to extractives during the winter would likely be more pronounced.

Here, we tested whether housing honey bees in hives constructed of western red cedar affects colony health compared to conventional pine boxes. In addition to overwintering survival, we also recorded colony strength metrics (adult bee frame coverage and brood area) pre- and post-wintering, as well as infestation of the major ectoparasite *Varroa destructor* [36]. To assess whether the bees may have been under sublethal stress, we evaluated the hemolymph proteome to quantify detoxification enzyme expression, among other stress indicators. This combination of colony-level metrics and protein expression data enabled us to assess the physiological health of the bees in addition to overall colony vitality. Because some beekeepers choose to use western red cedar boxes and report no adverse effects (anecdotally), we predicted that colony health metrics and overwintering survival would be similar between groups, but that detoxification enzyme levels may change.

## Methods

### Equipment and honey bee colonies

Five-frame hive boxes made of unstained, unpainted western red cedar and white pine were purchased from Urban Bee Supplies (Delta, BC). The cedar boxes (5.75 kg each) were originally sourced from a supplier on Vancouver Island and were constructed of old growth, passively dried (as opposed to kiln-dried) lumber. The lumber was allowed to dry for two months before constructing the beehives, and honey bees were introduced to the hive boxes three months later, when the wood was still highly aromatic. Since western red cedar extractives are typically most concentrated in old growth wood [16,20], and the known insecticidal compounds are volatile, this choice of wood and handling means that the material likely approached the upper limits of extractive content. The pine boxes (6.30 kg each) were originally sourced from Dancing Bee Equipment (Port Hope, ON).

Experimental nucleus colonies (n = 10 in cedar boxes and n = 10 in pine boxes) were produced from parent colonies originally established from New Zealand packages. On July 27th, parent colonies were sampled to determine the *Varroa* mite levels using the alcohol wash method [37], all of which were < 2%. The next day, nucleus colonies were produced, with each made up of two brood frames, two honey frames, one frame with space for the queen to lay, and one young mated queen received from a single batch imported from Olivarez Honey Bees (Orland, CA). Each parent colony produced at least two nucleus colonies with at least one in a cedar box and one in a pine box. Nucleus colonies were immediately transported to the experimental yard located ~ 25 km away and arranged in linear blocks of 4 colonies each (see Fig 1 for a schematic).

Four days later, colonies were inspected for queen acceptance and all queens were marked with a Posca paint pen. Two queens had damaged legs and were replaced with new Olivarez queens from the same cohort that had been held temporarily in a queen bank. On August 5th, colonies were again inspected and all queens were confirmed to have been accepted. Any colonies appearing to have fewer bees than the rest were bolstered with young bees obtained from larger donor colonies in order to normalize populations.

### Colony assessments and winter preparation

Photographs of both sides of each frame were taken on two occasions; once on September 1st, 2023 and again on April 1st, 2024. Each frame was labelled, placed on a vertical frame stand, and both sides were photographed. If adult bee coverage was too dense to clearly see the brood

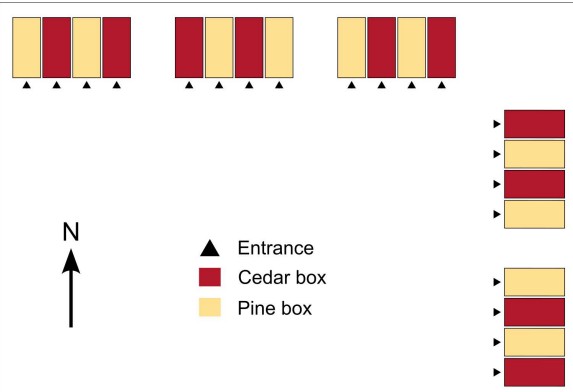

**Fig 1. Schematic of the experimental yard layout.** Nucleus colonies were arranged in four groups of five in alternating orders to minimize potential edge effects.

area, the bees were gently dispersed and a second set of photos were taken. To calculate adult bee coverage and brood area, images were analyzed in ImageJ (version 1.54d). First, total comb area (not including the outer wooden frame), in pixels, was calculated by setting polygons and using the "measure" function. Then, the adult bee area was traced and pixels were again measured. This was repeated to calculate the total brood area (including both open and capped brood), then both parameters were expressed as percent coverage relative to the comb area.

To prepare for winter, colonies were fed with 2:1 (sugar-to-water, w/w) syrup throughout September, and pre-winter colony weights (including the mass of the box) were recorded on October 3rd. These weights were then corrected for the average mass difference between the cedar and pine boxes (pine boxes were 0.55 kg heavier than cedar boxes). On September 25, one strip of Apivar was installed in each colony to control *Varroa*. On November 8th, the Apivar was removed and colonies were wrapped (four per bundle) in Reflectix® double-foil insulation and covered with a sheet of 2" thick polystyrene foam.

## Spring sampling

On the same day spring photos were taken (April 1st), foragers and newly emerged bees (10 pooled per colony) were sampled for proteomics analysis. Foragers were obtained by selecting bees walking on the frames that were carrying pollen loads. Newly emerged bees were obtained by selecting bees with soft bodies and a light grey appearance, indicating that they emerged within the previous 24 h. The bees were placed in 15 mL conical tubes and frozen immediately on dry ice, then stored at −70 °C until further processing. Once all molecular samples and photos were taken, a sample of bees ( ½ cup) was also obtained to determine mite infestation levels by the alcohol wash method. Spring mite treatments were not applied until after termination of the experiment.

## Proteomics sample preparation

We obtained hemolymph from foragers and newly emerged bees using a previously described method [38]. One hindleg was removed while frozen, then, 1 µl of the hemolymph exuding from the cavity where the coxa met the thorax was collected and deposited into a microfuge tube on ice that contained 10 µl of phosphate-buffered saline. Samples from bees belonging to the same colony were pooled in a single tube, then vortexed and a subsample (5 µl) was taken for proteomics sample preparation.

Protein samples were prepared for mass spectrometry exactly as previously described [39]. Briefly, protein was isolated via overnight acetone precipitation (80% final acetone concentration) and suspended in 20 μl urea digestion buffer (8 M urea, 2 M thiourea, 100 mM Tris, pH 8.0). Then, 10 μg of protein was reduced (0.2 μg dithiothreitol), alkylated (1 μg iodoacetamide), and digested (0.4 μg Lys-C/Trypsin mix, 3 h at room temperature followed by addition of 50 mM ammonium bicarbonate and overnight digestion). The next day, samples were acidified and desalted using C18 STAGE tips [40] as previously outlined [39]. Sample concentrations were determined using a NanoDrop (210 nm absorbance) and normalized to 10 ng/μl in buffer A (0.5% acetonitrile, 0.5% formic acid).

## Mass spectrometry analysis and data processing

For each sample, 50 ng of peptides were injected onto a NanoElute2 UHPLC system (Bruker Daltonics) with an Aurora Series Gen2 (CSI) analytical column (25 cm × 75 μm 1.6 μm FSC C18, with Gen2 nanoZero and CSI fitting; Ion Opticks) coupled to a timsTOF Pro2 mass spectrometer (Bruker Daltonics) operated in data independent acquisition-parallel accumulation serial fragmentation (DIA-PASEF) mode. The liquid chromatography method and the mass spectrometry data acquisition method was exactly as described for the *A. mellifera* samples in our previous publication [39].

Mass spectrometry data processing was conducted using DIA-NN [41] (version 1.8.1) with the library-free search method enabled (and a FASTA file was provided containing the *Apis mellifera* proteome downloaded from Uniprot in November, 2023, in addition to common honey bee intracellular pathogens and mass spectrometry contaminants list [42]). Search parameters were exactly as previously described [39], with peptide and protein identifications controlled to a 1% false discovery rate (FDR). All raw files, sample metadata, label-free quantitation results, and the FASTA search database are available on the MassIVE (https://massive.ucsd.edu/ProteoSAFe/static/massive.jsp) proteomics data archive under the accession MSV000093767.

## Statistical analysis

We conducted all statistical analyses in R (version 4.3.0) via R Studio (build 494) [43]. Colony metrics for cedar and pine colonies were compared using simple linear models, inspecting residual distributions to confirm appropriateness of fit. Differential expression analysis of proteomics data was performed using tools within the limma package [44]. Briefly, the normalized label-free quantitation data were first filtered to remove contaminant sequences and proteins identified in fewer than 75% of samples. Then, data were log2 transformed and examined for the presence of a significant interaction between hive type (two levels: cedar and pine) and bee age (two levels: newly emerged and forager), since it is possible that foragers that have spent their whole adult lives being exposed to their hive type may exhibit a stronger response than newly emerged bees that have only had the opportunity to contact wax comb during development. To do this, we fit a means model (using the lmFit function) with each group combination specified (newly emerged cedar, newly emerged pine, forager cedar, and forager pine) and analyzed the interaction (*i.e.*, the difference of differences) using the makeContrasts function followed by contrasts.fit and eBayes functions. When the interaction was determined not to be significantly influential (no proteins were differentially expressed), the contrasts were correspondingly adjusted to determine the effect of each factor individually. In all cases, the Benjamini-Hochberg correction was used to control the false discovery rate (FDR) to 5%.

GO term enrichment analysis of proteins differentially expressed between nurses and foragers was conducted using the gene score resampling approach within ErmineJ [45], with upregulated and downregulated proteins evaluated separately. Raw p values were used as the input "scores" and enrichment FDRs were controlled to 5% using the Benjamini-Hochberg method.

## Results

### Colony metrics

We found no significant differences between colonies housed in cedar versus pine boxes for any of the colony metrics measured at any time point (Fig 2, S1 File). Pre-winter colony weights were similar ($F_{1,18}$ = 0.39, p = 0.54) averaging at 21.5 kg for cedar hives and 21.8 kg for pine hives. Average adult bee frame coverage was similar in the two groups at the end of summer ($F_{1,18}$ = 2.2, p = 0.16) and spring ($F_{1,16}$ = 0.27, p = 0.97), as was the average brood area (end of summer: $F_{1,18}$ = 0.28, p = 0.60; spring: $F_{1,16}$ = 1.3, p = 0.73), despite one pine colony becoming broodless in September during queen supersedure. Identical, high overwintering survival was achieved in both groups (90%), with the supersedure queen being among those that survived. In April, *Varroa* mite levels were relatively high in both groups (averaging at 2.9% in cedar boxes and 2.6% in pine boxes) but were statistically indistinguishable ($F_{1,16}$ = 0.13, p = 0.73).

### Protein expression

Proteomic analysis of day-old bee and forager hemolymph shows that, although 4,674 protein groups were identified (of which 3,146 were considered quantified after filtering) and 2,612 were differentially expressed among day-old bees and forager bees (including expected patterns for vitellogenin; Fig 3A and S2 File), no proteins significantly varied according to box type when that factor was considered as either an interacting or a non-interacting factor. Sixteen detoxification enzymes were quantified, with none of them linked to box type, while fourteen were differentially expressed according to bee age (Fig 3B).

We found no differences in expression patterns of canonical stress response proteins (heat-shock proteins (HSPs), immune factors, and antioxidant enzymes) with regard to hive type, but strong differences were observed according to worker age (Fig 4A). We see pronounced

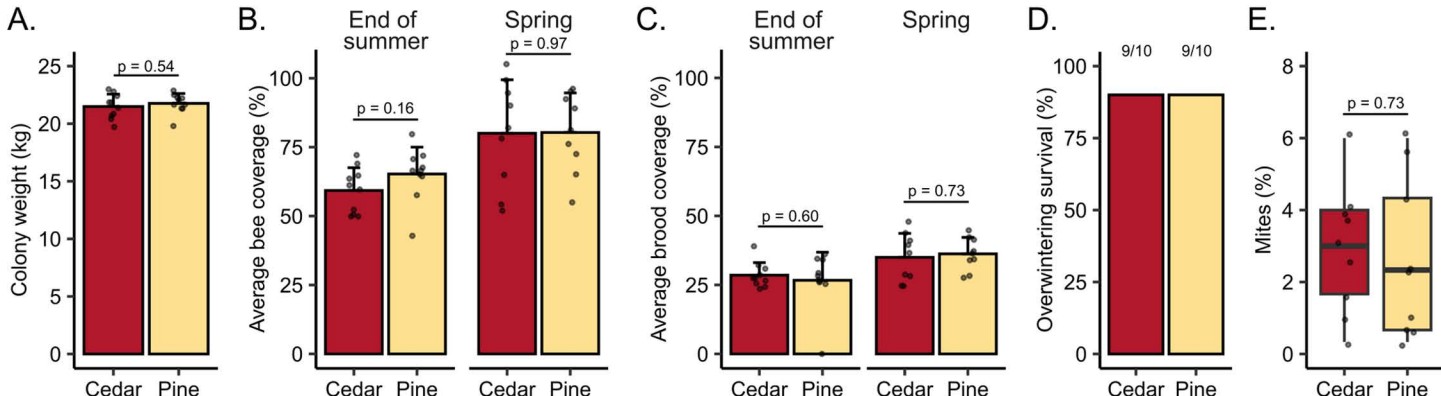

**Fig 2. Hive wood type does not affect colony phenotypes.** A) Pre-winter colony weights of the 5-frame nucleus colonies were measured on October 3rd, and the average weight of the nuc boxes were subtracted. B) Adult bee coverage (percent of frame area occupied) and C) brood area was measured at the end of summer (September 1st) and again in the spring (April 1st). D) Ninety percent of colonies (9/10) survived the winter in each group. E) Spring mite loads were similar between groups.

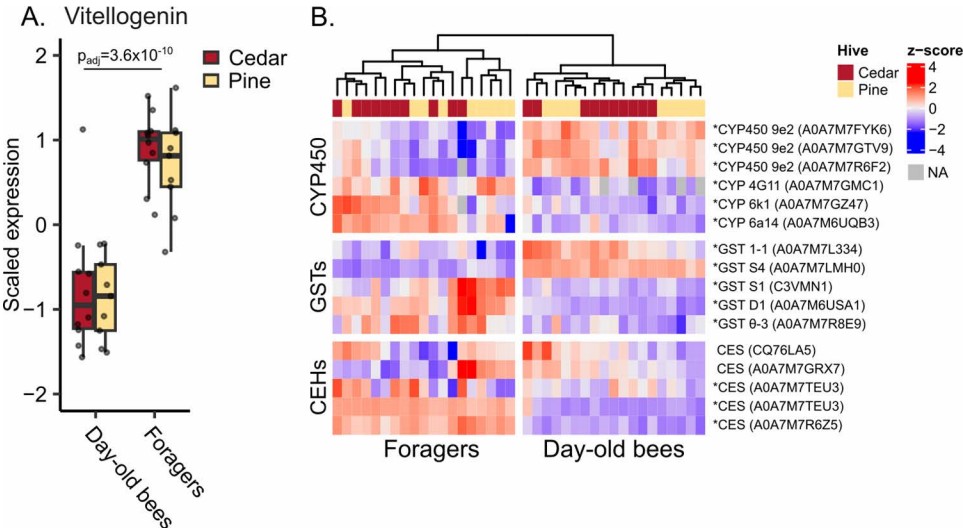

**Fig 3. Hemolymph detoxification enzyme expression does not correlate with wood type.** A) Vitellogenin exhibits the expected expression patterns according to worker task, but is not impacted by box wood type. B) Cytochrome p450 (CYP450) enzymes, glutathione-S-transferases (GSTs), and carboxylesterases (CESs) were differentially expressed according to worker task (with the exception of two CESs), but not hive wood type. Asterisks indicate significance according to worker task (5% FDR, Benjamini-Hochberg correction). NA = not identified.

patterns of differential expression in all three groups of proteins, with day-old bees expressing high levels of most, but not all HSPs, among which only a subset of four proteins (one DNAJ and three members of the protein lethal 2 essential for life gene family, or pl(2)el) were upregulated in foragers. In contrast, all immune proteins and almost all antioxidant enzymes inspected (except one glutathione peroxidase) were upregulated in foragers relative to day-old bees. Thousands of other proteins were differentially expressed in addition to those shown here, and all can be found in S2 Data. Among proteins upregulated in foragers, six GO terms were enriched (5% FDR), all of which were related to metabolic function (Fig 4B). Among proteins upregulated in day-old bees, 11 GO terms were enriched, with functions related to protein folding (a consequence of the strong HSP expression), translation, and structural components of vesicles (Fig 4C).

## Discussion

We identified no significant effect of wood type on any of the variables we evaluated, whether at the colony or physiological level. This is despite designing our experiment in a way that should achieve a high level of exposure: 1) We used cedar lumber derived from old growth trees (which typically have higher extractive concentrations [16,20]), 2) the lumber was air dried (not kiln-dried) and minimally aged, and 3) we hived the bees over winter, when their long lifespan [35,46] and increased time spent inside the hive [33] should maximize exposure. However, since we did not quantify the levels of known insecticidal compounds within the cedar wood, we cannot comment on the specific intensity of exposure that the bees experienced, nor did we measure an exhaustive list of outcomes (impacts of wood type on behaviors and hive matrices, for example, were not recorded). Nevertheless, since we observed no discernable effect among the parameters we did measure, we conclude that western red cedar is likely an innocuous hive material.

We originally predicted that we would identify no effects of wood type on colony metrics, but positive or negative effects could also have been conceivable. Extracts from other cedar

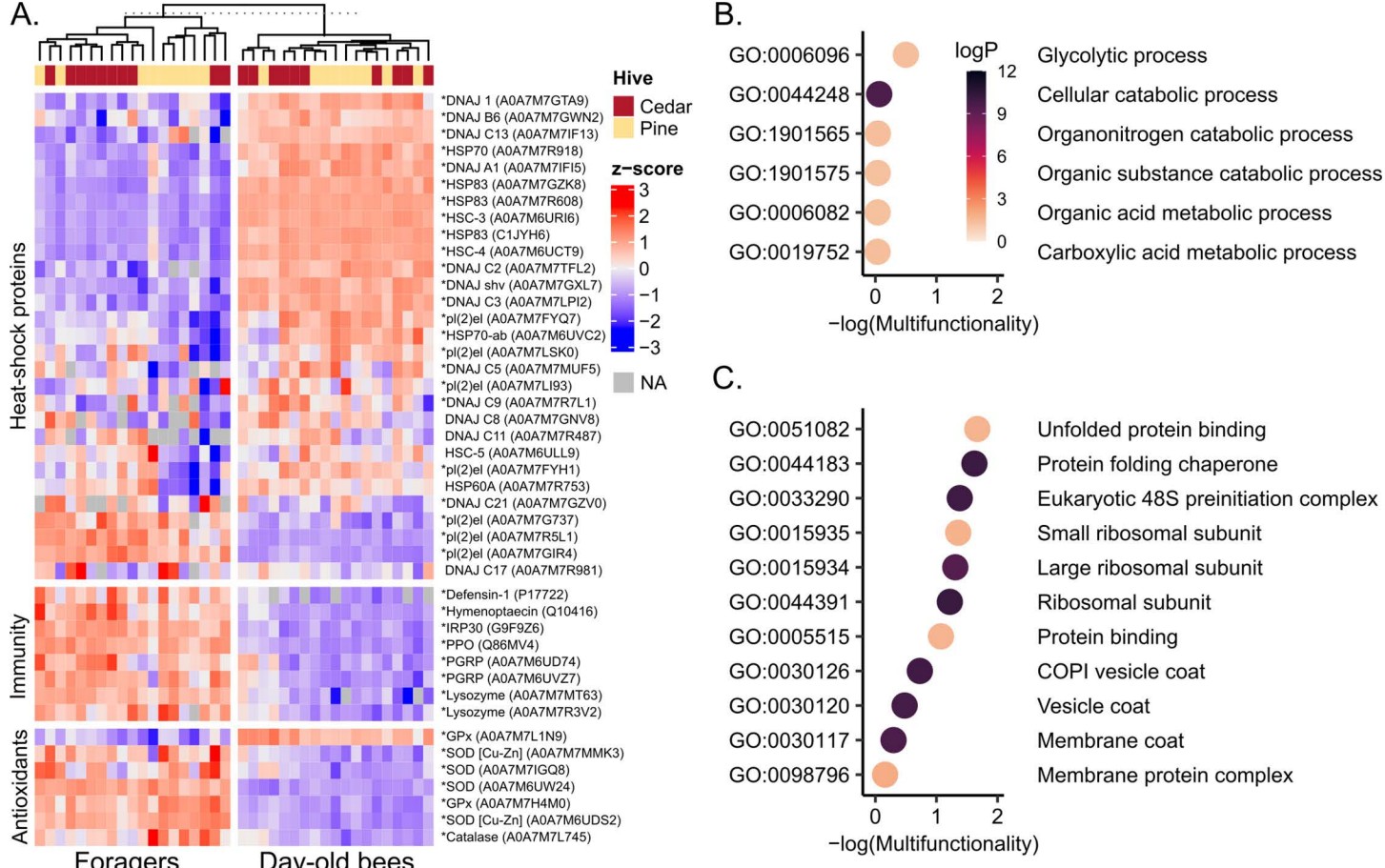

**Fig 4. Stress response proteins are strongly differentially expressed by worker task. A**) All quantified heat-shock proteins, immune proteins, and antioxidant enzymes. Asterisks indicate significance with respect to worker task (5% FDR, Benjamini-Hochberg correction). No proteins were significant with respect to hive wood type. B) GO terms significantly enriched for proteins upregulated in foragers and C) downregulated in foragers (5% FDR, Benjamini-Hochberg correction). GO terms with higher -log(multifunctionality) scores are less likely to be enriched as a result of protein multifunctionality.

species are toxic to at least two species of tick (which belongs to the same taxonomic class as the *Varroa* mite, a major honey bee pest [36]) [7,11]. We therefore speculated that we might observe a negative effect on spring *Varroa* infestation levels, but we found that infestations were similar between groups at 2.9% (cedar boxes) and 2.6% (pine boxes) (Fig 1E). Moreover, western red cedar is also a better insulator than pine [16]; a particularly useful characteristic during the cold winter temperatures. Despite seeing no impact on colony outcomes (Fig 1), this could still be a beneficial feature, but with an effect overcome by the fact that we wrapped colonies in rows that allowed them to share heat. Additional testing using independent, non-insulated units would be necessary to determine if the insulating properties provide a practical benefit.

While we expected to see largely null effects of hive type on colony metrics, we were less certain about the impact on detoxification enzyme expression. Sublethal effects of pesticides, including gene expression changes, have been innumerably documented in honey bees in other contexts (reviewed in [47,48,49]). The fact that we quantified sixteen detoxification enzymes belonging to all major functional classes and none of them varied according to hive type (Fig 3B) suggests that western red cedar is likely physiologically benign, in addition to having no effect on overall colony performance.

We did, however, detect extensive protein expression differences among newly-emerged bees and foragers, which are known to be physiologically distinct [33]. This shows that we did not simply fail to identify differences according to hive type due to a lack of quantitative power or poor data quality. Specifically, we show that vitellogenin was less abundant in newly-emerged bees compared to foragers (Fig 3A), which agrees with previous research showing that in abdomens, vitellogenin expression is lowest in newly-emerged bees compared to other adult life stages [50]. Other expression patterns also agree with published data, namely, that all canonical immune proteins were expressed at low levels in newly-emerged bees (Fig 4A), which have not yet activated their immune system [51,52].

The expected patterns for antioxidant enzymes and HSPs were less clear, as newly-emerged bees are not often evaluated in expression studies. One might hypothesize that foragers would also express higher levels of these proteins, since their increased metabolic rate makes them more prone to oxidative stress [53], and they are presumably more likely to be exposed to extreme temperatures while travelling outside of the hive. The former hypothesis was largely supported (with the exception of one glutathione peroxidase, A0M7A7L1N9, that was expressed more strongly in newly-emerged bees) while the latter was not. Interestingly, foragers largely exhibited low expression levels of HSPs and related proteins, with the exception of one DNAJ protein (A0A7M7GZV0) and three members of the pl(2)el family (small HSPs; A0A7M7G737, A0A7M7R5L1, and A0A7M7GIR4). Interestingly, other members of the pl(2)el family were expressed more strongly in newly-emerged bees. Members of this gene family respond differently to distinct stressors [54], and our data clearly show they also have distinct age-dependent expression patterns.

Although we report only null effects of hive type and conclude that western red cedar is likely a suitable construction material for beehives, we acknowledge that there may be yet untested contexts in which this is not the case. For example, extractive content can vary from batch to batch of lumber, and while we utilized hives derived from old growth trees (which have, on average, a higher extractive concentration [16,20]), we did not actually analyze the concentrations of bioactive compounds in the wood. There is a wide spectrum of potential concentrations even among old growth wood, so it is possible that the exposure levels in our study were not sufficient to observe an effect. While our data do not reveal any positive or negative effects of western red cedar, there are many species of trees referred to as "cedar," and each one should be tested to verify that it is benign (or beneficial) before being used on a large scale.

## Supporting Information

**S1 File. Colony metrics.** All data describing colony phenotypes (pre-winter weights, brood area, adult bee coverage, survival, and spring mite infestation).
(XLSX)

**S2 File. Proteomic data.** All data and analyses derived from the protein expression data (protein expression matrix, sample metadata, differential expression analysis, and GO term enrichment analysis).
(XLSX)

## Acknowledgements

We would like to acknowledge Jason Rogalski, Jeanne Yuan, and Renata Moravcova for their expertise and assistance with generating the proteomics data, as well as the beekeepers and equipment suppliers we consulted while designing this experiment, and the land owner who allowed us to conduct this research on their property.

## Author contributions

**Conceptualization:** Alison McAfee, David R Tarpy, Leonard J Foster.

**Data curation:** Alison McAfee.

**Formal analysis:** Alison McAfee.

**Funding acquisition:** Alison McAfee, Leonard J Foster.

**Investigation:** Alison McAfee.

**Methodology:** Alison McAfee.

**Visualization:** Alison McAfee.

**Writing – original draft:** Alison McAfee.

**Writing – review & editing:** David R Tarpy, Leonard J Foster.

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
