## [Decision Letter · Decision Letter 0]

5 Feb 2025

PONE-D-25-03510Western red cedar (Thuja plicata) beehives have no impact on honey bee (Apis mellifera) overwintering colony survival or detoxification enzyme expressionPLOS ONE

Dear Dr. McAfee,

Thank you for submitting your manuscript to PLOS ONE. After careful consideration, we feel that it has merit but does not fully meet PLOS ONE’s publication criteria as it currently stands. Therefore, we invite you to submit a revised version of the manuscript that addresses the points raised during the review process.

We look forward to receiving your revised manuscript.

Kind regards,

Yahya Ahmed Shaban Al Naggar

Academic Editor

PLOS ONE

Journal Requirements:

2. Please ensure that the title is match in the manuscript and the pdf file.

This work was supported by a grant from the Canadian Bee Research Fund to A.M. and L.J.F. (no applicable grant number) and a grant from the Natural Sciences and Engineering Research Council (RGPIN-2022-03022) to L.J.F. Mass spectrometry infrastructure used here was supported by grants from PacifiCan (grant number 22637), the Canada Foundation for Innovation and the BC Knowledge Development Fund (grant number 43403 for both), and the University of British Columbia Life Sciences Institute (no applicable grant number). 

4. Please remove all personal information, ensure that the data shared are in accordance with participant consent, and re-upload a fully anonymized data set. 

Reviewers' comments:

Reviewer's Responses to Questions

**Comments to the Author**

1. Is the manuscript technically sound, and do the data support the conclusions?

Reviewer #1: Yes

Reviewer #2: Yes

2. Has the statistical analysis been performed appropriately and rigorously? 

Reviewer #1: Yes

Reviewer #2: Yes

3. Have the authors made all data underlying the findings in their manuscript fully available?

Reviewer #1: Yes

Reviewer #2: Yes

4. Is the manuscript presented in an intelligible fashion and written in standard English?

Reviewer #1: Yes

Reviewer #2: Yes

5. Review Comments to the Author

Reviewer #1: I always appreciate practically oriented articles, and your contribution is both easy to read and highly understandable. The design of your experiment is well-structured. While there are other applicable physiological indicators for measuring colony stress, I believe that evaluating detoxification enzyme expression is sufficient for a pilot study. Congratulations on presenting an inspiring idea worth pursuing (it would be valuable to test other resin-rich woods, including some types of pine) and for delivering a concise and focused analysis without delving unnecessarily into peripheral topics.

Reviewer #2: This manuscript presents a well-designed and executed study investigating the effects of western red cedar hive boxes on honey bees. The authors find no evidence of adverse effects at both the colony and molecular levels. While further research is a demand, the study provides valuable preliminary data suggesting that western red cedar is likely a safe material for hives especially that most of the researches used extracts from the cedar wood not the wood itself. The manuscript is suitable for publication after addressing the minor revisions that will be suggested.

1- While Authors selected old-growth, air-dried cedar, they didn't quantify the actual concentration of extractives in the wood. This makes it difficult to determine the true level of exposure. Analyzing the chemical composition of the wood used would strengthen the conclusions.

2- Why did not the authors analyzed the stored bee bread, wax combs or honey from experimental colonies as they would be very sensitive to any volatile components.

3- Is the Western red cedar (Thuja plicata) wood is the only type of cedar that is used in North America to manufacture hives.

4- As most volatile components are affected by the time authors should specify the period of cedar wood storage (as row material and hives) before using in the experiment.

5- As mentioned before that cedar hives are better isolated, then the honey area in combs should have been measured to ensure that.

6- if there any behavioral changes in the colonies, it should be mentioned especially in the forage and workers policing

6. PLOS authors have the option to publish the peer review history of their article (what does this mean? ). If published, this will include your full peer review and any attached files.

**Do you want your identity to be public for this peer review?** For information about this choice, including consent withdrawal, please see our Privacy Policy .

Reviewer #1: **Yes: ** Robert Chlebo

Reviewer #2: **Yes: ** Hatem Sharaf El-Din

---

## [Author Response · Author response to Decision Letter 1]

18 Feb 2025

Please see the attached response to reviewers documented for changes made in light of all reviewer and editorial requests

---

## [Editor Report · Decision Letter 1]

23 Feb 2025

Western red cedar (*Thuja plicata* ) beehives have no impact on honey bee (*Apis mellifera* ) overwintering colony survival or detoxification enzyme expression

PONE-D-25-03510R1

Dear Dr. McAfee,

We’re pleased to inform you that your manuscript has been judged scientifically suitable for publication and will be formally accepted for publication once it meets all outstanding technical requirements.

Within one week, you’ll receive an e-mail detailing the required amendments. When these have been addressed, you’ll receive a formal acceptance letter, and your manuscript will be scheduled for publication.

If your institution or institutions have a press office, please notify them about your upcoming paper to help maximize its impact. If they’ll be preparing press materials, please inform our press team as soon as possible—no later than 48 hours after receiving the formal acceptance. Your manuscript will remain under strict press embargo until 2 pm Eastern Time on the date of publication. For more information, please contact onepress@plos.org.

Kind regards,

Yahya  Al Naggar

Academic Editor

PLOS ONE
---

## [Editor Report · Acceptance letter]

PONE-D-25-03510R1

PLOS ONE

Dear Dr. McAfee,

I'm pleased to inform you that your manuscript has been deemed suitable for publication in PLOS ONE. Congratulations! Your manuscript is now being handed over to our production team.

Kind regards,

on behalf of

Dr. Yahya Al Naggar

Academic Editor

PLOS ONE